# A d^10^-Cd Cluster Containing Sandwich-Type Arsenotungstate Exhibiting Fluorescent Recognition of Carcinogenic Dye in Methanol

**DOI:** 10.3390/molecules29215193

**Published:** 2024-11-02

**Authors:** Feng Wang, Xiang Ma, Haodong Li, Ziqi Zhao, Lele Zhang, Yutong Zhao, Haipeng Su, Zeqi Wang, Changchun Li, Jiai Hua

**Affiliations:** 1Chemistry and Chemical Engineering Department, Taiyuan Institute of Technology, Taiyuan 030008, China; wangf@tit.edu.cn (F.W.); 16603265568@163.com (H.L.); 16613802856@163.com (L.Z.); 19581525302@163.com (Y.Z.); wzq15003489249@163.com (Z.W.); 2Laboratory of Biochemistry and Pharmacy, Taiyuan Institute of Technology, Taiyuan 030008, China; 13394412558@163.com (Z.Z.); 19935347213@163.com (H.S.); l17850742078@163.com (C.L.)

**Keywords:** Cd cluster, polyoxometalate, fluorescence recognition, carcinogenic polycyclic aromatic hydrocarbons

## Abstract

A d^10^-Cd cluster containing sandwich-type arsenotungstate [C_3_H_12_N_2_]_6_[Cd_4_Cl_2_(B-*α*-AsW_9_O_34_)_2_] was synthesized and its structure characterized through elemental analyses, X-ray powder diffraction (XRPD), IR spectroscopy, X-ray photoelectron spectroscopy (XPS), and single-crystal X-ray diffraction. The X-ray analysis revealed that the molecular unit of the compound consists of a captivating tetra-Cd-substituted sandwich-type polyoxoanion, accompanied by six elegantly protonated 1,2-diaminopropane as counter ions. The further novelty of the tetranuclear cadmium cluster lies in its occupied chlorine atom sites. This makes it highly susceptible to coordinate reactions with nitrogen on polycyclic aromatic hydrocarbons, thereby exhibiting different fluorescent signals that facilitate the identification and detection of these carcinogenic substances in methanol.

## 1. Introduction

Dye molecules are essential in daily life; however, the impact of environmental pollution cannot be disregarded when considering their commercial value [1]. Dyes typically contain various carcinogenic pollutants, such as aniline, dimethylamine hydrochloride, ortho-toluidine, 4,4′-diaminodiphenylmethane, cyclohexylamine nitrate, etc. [2]. These substances are widely used in industries including printing, rubber production, plastic manufacturing, leather processing, and pesticide production [2]. The residues of these dyes in manufactured products are frequently resistant and possess a potent carcinogenic impact, and the development of efficient technologies or systems for promptly detecting these harmful substances is imperative to mitigate the impacts [3]. However, conventional detection technologies, such as colorimetry, spectrophotometry, atomic absorption spectroscopy, electrochemical analysis, and gas chromatography, suffer from various limitations including the requirement for complex instrumentation, sophisticated procedures/sampling techniques, high cost implications, and intricate sample preparation protocols [4,5,6]. Therefore, it is crucial and imperative to develop sensitive, selective, rapid, and cost-effective analytical methods. Advanced sensing materials enable rapid identification of hazardous/toxic species, facilitating efficient detection and subsequent sequestration and removal of these pollutants [6]. Recently, fluorescent sensors have garnered significant attention due to their capacity for providing a straightforward, highly sensitive, selective, precise, and cost-effective approach for real-time monitoring without necessitating any sample pretreatment, alongside their advantageous spatial and temporal resolution [6,7].

Metal atoms with a d^10^ electron configuration, such as zinc ions and cadmium ions, exhibit excellent fluorescence properties, and the coordination polymers (CPs) based on those ions have found extensive applications in the recognition of specific small molecules, showcasing a remarkable level of advancement [7,8,9,10]. Therefore, theoretically, the clustering of Cd^2+^ should result in enhanced fluorescence intensity and increased availability of fluorescence information.

Polyoxometalates (POMs), being an intriguing category of metal–oxygen clusters, exhibit diverse structures and captivating properties across various domains [11,12]. POMs possess several inherent and irreplaceable advantages, including a nucleophilic oxygen-enriched surface, nano-size dimensions, and poly-bond-making sites [13,14,15,16,17]. These features enable POMs to function as bulky polydentate ligands that can incorporate transition metal ions with flexible coordination modes [18,19,20,21,22], in which lacunary polyoxometalates (POMs) not only facilitate the formation but also sustain the high nuclearity of transition metal clusters [23]. Although there have been reports on high-nuclearity POMs of first transition metal clusters [24], they are predominantly concentrated in the field of magnetic research [25]. However, there is a scarcity of reported high nuclear Cd-cluster-substituted POMs with fluorescence activity, and the investigation into Cd^2+^ clusters in this aspect is particularly lacking [26].

In this paper, we report a d^10^-Cd cluster containing sandwich-type arsenotungstate, (H_2_dap)_6_[Cd_4_Cl_2_(B-*α*-AsW_9_O_34_)_2_]·8H_2_O (dap = 1,2-diaminopropane, abbreviated as Compound **I**, Figure 1), which was derived from a pure inorganic {Cd_4_} cluster sandwiched by two trivacant Keggin-type tungstoarsenates [B-*α*-AsW_9_O_34_]^9–^. To our knowledge, **I** represents a rare tetra-Cd cluster-substituted sandwich-type POM. Moreover, it can recognize typical carcinogenic dye in methanol through unique fluorescent spectra.

## 2. Results

### 2.1. X-Ray Single-Crystal Structures

The structure of **I** was determined through single-crystal X-ray diffraction analysis, providing valuable crystallographic data and selected bond lengths that are summarized in Appendix A. The detailed information has been submitted to the Cambridge Crystallographic Data Center and assigned a CCDC number of 2341526. The X-ray structural analysis unveils that the molecular structural unit of **I** comprises a captivating tetra-Cd-substituted sandwich-type polyoxoanion [Cd_4_Cl_2_(B-*α*-AsW_9_O_34_)_2_]^12–^, accompanied by six elegantly protonated 1,2-diaminopropane (H_2_dap)^2+^ as counter ions and eight meticulously arranged crystallization water molecules. The polyoxoanion, as depicted in Figure 1A, consists of a tetra-Cd cluster {Cd_4_Cl_2_O_14_} in the shape of a parallelepiped (abbreviated as {Cd_4_}, Figure 1B) sandwiched between two trivacant Keggin-type [B-*α*-AsW_9_O_34_]^9–^ fragments (Figure 1C) in a staggered conformation with *Ci* symmetry. The formation of [B-*α*-AsW_9_O_34_]^9–^ fragments can be inferred from the presence of Figure 1D, where it is evident that these fragments are generated by removing three edge-sharing WO_6_ octahedra from a saturated Keggin-type cluster belonging to one edge-sharing W_3_O_13_ triad. The tetra-Cd cluster in Figure 1B is connected by two [B-*α*-AsW_9_O_34_]^9–^ units through the participation of 14 O*_μ_* from their lacunae, including two O*_μ_*_4_ from {AsO_4_} tetrahedra and 12 O*_μ_*_2_ from 12 {WO_6_} octahedra. The coordination environments of four Cd^2+^ ions can be classified into two types: Cd1 adopts a distorted octahedral geometry, which is defined by one Cl atom with a Cd–Cl bond length of 2.511 Å, and five O atoms from two [B-*α*-AsW_9_O_34_]^9–^ fragments with Cd-O bond lengths ranging from 2.223 to 2.512 Å. The Cd2 ions adopt octahedral coordination configurations with Cd2–O bond distances ranging from 2.165 to 2.341 Å, occupying the four vacant sites of two [B-*α*-AsW_9_O_34_]^9–^ fragments to form sandwich-type [Cd_2_(B-*α*-AsW_9_O_34_)_2_]^14–^ moieties. The structural data suggest that only two sites, where Cl atoms are located, can interact with residues of those toxic dyes.

### 2.2. XRPD, IR Spectrum, UV–Vis Spectra, and TGA

The IR spectrum of **I** exhibits analogous asymmetric vibrations to other species containing [B-*α*-AsW_9_O_34_]^9–^ [27]. The vibrational spectra in Figure 2A exhibit three distinct bands corresponding to the *v*(W–O_t_), *v*(W–O_μ2_), and *v*(W–O_μ3_) modes, which manifest at 940, 783, and 733–625 cm^–1^, respectively [27]. The vibrational mode at 874 cm^–1^ is ascribed to the stretching of the As–O*_μ_*_4_ bond [27]. The presence of dap is confirmed by the characteristic bands at 3410 cm^–1^, which are attributed to stretching vibrations of N–H. The vibration observed at 1614 cm^–1^ can be attributed to the stretching of the hydroxyl (–OH) group. The IR spectrum exhibits excellent concordance with the findings of X-ray diffraction structural analysis. The phase purity of **I** was confirmed by juxtaposing the experimental X-ray powder diffraction (XRPD) pattern with the simulated pattern derived from single-crystal X-ray diffraction (Figure 2B).

The UV spectrum in aqueous solution, as depicted in Figure 2C, exhibits two absorption peaks: one at 193.8 nm and the other characterized by a broad shoulder absorption at 257.5 nm within the range of 190 to 600 nm. The assignment of these two peaks can be attributed to the charge transfer transitions from O*_t_* to W and O*_μ_* to W, respectively [28]. The impact of pH value on the stability of **I** was also explored through UV–visible spectra. The UV–visible absorption peaks of **I** exhibited no significant variation in intensity between pH 4.50 and 8.00, as depicted in the inset of Figure 2C. The absorption peaks at both 193.8 nm gradually alter in intensity beyond the aforementioned pH range, potentially suggesting the gradual disintegration of **I**’s framework.

The thermogravimetric (TG) curve of **I**, as depicted in Figure 2D, exhibits a dual-stage weight loss phenomenon, resulting in an overall reduction of 19.06% within the temperature range of 25–600 °C. The initial stage, ranging from 40 to 130 °C, is ascribed to the evaporation of eight lattice water molecules. The observed weight reduction of 2.90% harmonizes precisely with the calculated value of 2.86%. The second stage, characterized by a weight loss of 16.16%, occurs within the temperature range of 190 to 566 °C. This phenomenon can be attributed to the elimination of six molecules of 1,2-dap, one As_2_O_3_ molecule, and four structural water molecules (calculated as 16.10%) [29].

### 2.3. Fluorescence Properties

The construction of coordination complexes using d^10^-metal ions and conjugated ligands have been reported to exhibit exceptional fluorescence properties, rendering them promising candidates for potential luminescent materials [30]. The intense emission bands were observed at *λ*_em_ = 395 and 376 nm (*λ*_ex_ = 205 nm) for **I**, as depicted in Figure 3. These bands can be attributed to the d→d transition [7].

Intermediates utilized in the synthesis of dyes frequently encompass carcinogenic substances and have the potential to endure within the final products, with benzidine, dimethylamine hydrochloride, *o*-tolidine, 4,4′-diamino-diphenylmethane, and dicyclohexylamine nitrite being prevalent intermediate carcinogens [31]. As the majority of them exhibit solubility in methanol, **I** molecules displaying significant fluorescence response in methanol have the potential to serve as effective probes for detecting these carcinogens. The samples containing carcinogens and their analogues in this test are showcased in Figure 4 because of their ubiquitous presence in industrial products, with their concentration meticulously designed to adhere to the stringent Chinese national standards, ensuring a minimum value of 20 mg/L.

Firstly, the initial step entails examining the fluorescence determination of naphthalene and its derivatives. As shown in Figure 5, following the incubation of **I**, the fluorescence signal of the selected naphthalene derivative exhibits a remarkable enhancement compared to its initial state. In this group, the fluorescence intensity of 1-naphthylamine (Figure 5A) exhibits a relatively superior magnitude in comparison to N-(1-naphthyl) ethylenediamine (Figure 5B). The potential explanation for this phenomenon lies in the enhanced efficiency of electron transfer from L to M subsequent to 1-naphthylamine’s coordination with Cd clusters compared to that of N-(1-naphthyl) ethylenediamine. Furthermore, the fluorescence peak of naphazoline (Figure 5C) and I not only intensifies after incubation but also exhibits the emergence of multiple distinctive fluorescence peaks, which may be attributed to the fact that naphazoline has the ability to exist in various coordination modes alongside Cd clusters.

Secondly, the fluorescence detection results of bipyridine and its analogues were tested. As shown in Figure 6, after incubation with **I**, derivatives of pyridine showed a significant increase in fluorescence signal with the exception for 2,2′-biisonicontinic acid. The remarkable enhancement in fluorescence signals of 2,2′-bipyridine (Figure 6A), 4,4′-bipyridine (Figure 6B), Phen (Figure 6D), and 4-(4-pyridinyl)thiazole-2thiol (Figure 6E) along with the emergence of distinctive peaks suggests their potential formation of intricate complexes with cadmium ions in diverse coordination configurations. The fluorescence of 2,2′-biisonicontinic acid (Figure 6C), however, remained unaltered in comparison to its previous state, thereby suggesting that its expanded molecular volume impeded its coordination with **I**.

Thirdly, the primary emphasis of the group lies in the exquisite fluorescence detection of **I** utilizing benzidine or aniline and their various derivatives. As shown in Figure 7, the fluorescence signal changes of structurally similar benzidines such as 2-aminobiphenyl (Figure 7A), 4-aminobiphenyl (Figure 7B), benzidine (Figure 7C), and *o*-tolidine (Figure 7D) exhibit a consistent trend with only minor nuances in details. Interestingly, the fluorescence signal of 3,3-diaminobenzidine (Figure 7E) remained unaltered both prior to and subsequent to incubation with **I**, suggesting a plausible absence of interaction between 3,3-diaminobenzidine and **I**. As shown in Figure 7F,G, the intriguing aspect lies in our remarkable ability to discern the fluorescence of benzene amine derivatives 4,4′-diaminodiphenylmethane and 3,5-dimethylaniline, with 3,5-dimethylaniline being particularly distinguished by its easily identifiable fingerprint peaks.

## 3. Discussion

The sandwich-type POM compounds were previously confined to the first transition series elements for the sandwich layer ion [32]. It is exceptionally rare for cadmium, a second transition series element, to form a tetranuclear-Cd cluster. According to hard-soft acid-base theory, Cd is a soft acid that readily forms complexes with ligands containing N, S, etc., which are classified as soft bases. Hence, the coordination of nitrogen in polycyclic aromatic hydrocarbons with Cd ions is advantageous, as it effectively enhances the fluorescence signal of Cd clusters by providing electron-donating groups [33]. Furthermore, compared to previous MOF materials having fluorescence recognition characteristics [34], another novel characteristic of **I** is that the selection of Cl ions as leaving groups not only reduces production costs but also enhances the sensitivity of compound fluorescence recognition, thereby bestowing a touch of elegance upon the overall process.

By virtue of its d^10^ electron configuration, Cd^2+^ inherently possesses captivating fluorescent characteristics [7]. The inclusion of tetra-Cd clusters further expands the realm of possibilities for d→d electron transfer, endowing **I** with a more opulent fluorescence signal characteristics. After coordination with benzene or pyridine rings possessing electron-donating ability, the electronic transition of Cd is modified, thereby amplifying its fluorescence signal intensity. Additionally, owing to the presence of dual coordination sites within cadmium clusters and diverse ligand coordination modes, it becomes feasible to generate complexes with varied configurations post-interaction, thereby augmenting the intricacy of fluorescence signals and engendering distinctive peaks akin to fingerprint characteristics of specific substances. In addition, due to the steric hindrance effect of POM’s building blocks, some molecules with large substituents are difficult to combine with cadmium clusters, which facilitates the discrimination between substituted and unsubstituted polycyclic aromatic compounds by amplifying the fluorescence signals.

## 4. Materials and Methods

### 4.1. Materials and Equipment

Reagents employed in this investigation were of analytical grade, procured from commercial vendors and utilized without further treatment unless otherwise specified. Methanol (MeOH), ethanol (EtOH), acetone, tetrahydrofuran (THF), 1,2-ethanediol, acetonitrile (CH_3_CN), N-methylpyrrolidone (NMP), dichloromethane (CH_2_Cl_2_), chloroform (CHCl_3_), N, N-dimethyl formamide (DMF), N, N-dimethyl acetamide (DMA), isopropanol (2-PA), CdCl_2_, NaAsO_2_, and Na_2_WO_4_·2H_2_O were purchased from J & K Scientific Inc. (P. R. China). 2-aminobiphenyl, 4-aminobiphenyl, benzidine, o-tolidine, 3,3-diaminobenzidine, 4,4-diaminodiphenylmethane, 3,5-dimethylaniline, 2,2′-bipyridine, 4,4′-bipyridine, 2,2′-biisonicotinic acid, 1,10-phenanthroline, 4-(4-pyridinyl)thiazole-2-thiol, 1-naphthylamine, N-(1-nphthyl)ethylenediamine, naphazoline, and 1,4-naphthalenedicarboxylic acid were purchased from Macklin Reagent Inc. (P. R. China). The solutions were all prepared using ultrapure water filtered through the Milli-Q academic system.

The X-Ray powder diffraction (XRPD) measurements were conducted at 293 K using a Philips X’pert-MPD instrument equipped with Cu-K*α* radiation (λ = 1.54056 Å). The IR spectrum was acquired from a sample powder that was either pelletized with KBr or dissolved in chloroform, using a Nicolet 170 SXFT-IR spectrophotometer spanning the range of 4000–400 cm^–1^. The single crystal data of compound **I** were collected using a Bruker CCD, Apex-II diffractometer equipped with graphite-monochromated Mo K*α* (λ = 0.71073 Å) radiation at ambient temperature. The routine Lorentz and polarization corrections were applied, followed by an absorption correction using the SADABS program. The crystal structure was determined using direct methods and refined through full-matrix least squares refinement based on *F^2^*. The SHELXL-97 program package was utilized for all calculations. The PQEXCeII ICP-MS instrument was utilized for elemental analysis.

### 4.2. Synthesis of Compound I

Two separate solutions were prepared. Solution A: Na_2_WO_4_·2H_2_O (3.299 g, 10.00 mmol) and NaAsO_2_ (0.430 g, 3.00 mmol) were dissolved in 100 mL of water with continuous stirring. Solution B was prepared by adding CdCl_2_ (0.916 g, 5.00 mmol) and 1,2-dap (0.06 mL, 0.850 mmol) to a stirred mixture of water (100 mL). The resulting mixture of B was added to solution A, followed by stirring for 10 min at room temperature. Subsequently, the pH value was adjusted to 5.50 through gradual addition of 6 mol·L^–1^ HCl. The solution was enclosed within a 25 mL Teflon-lined autoclave, which was maintained at a temperature of 160 °C for a duration of 3 days, followed by gradual cooling to reach room temperature. The transparent rhomboid crystals were collected, yielding 0.61 g (approximately 22% based on Na_2_WO_4_·2H_2_O).

### 4.3. X-Ray Data Collection and Structure Refinement

The intensity data were collected using a Bruker APEX-II CCD diffractometer equipped with Mo K*α* monochromated radiation (λ = 0.71073 Å) at a temperature of 296(2) K. The intensities were corrected using Lorentz–polarization factors and multi-scan absorption techniques. The crystal structure was first solved with the ShelXT structure solution program by the utilization of direct methods and then refined with the ShelXL-2018/3 package [35]. The integration of data was conducted utilizing the SAINT method [36]. The routine Lorentz and polarization corrections were applied, while the multi-scan absorption corrections were performed using SADABS. A summary of crystallographic data and structure refinement for compound **I** is presented in Appendix A. The bond lengths and angles of **I** are documented in Appendix A.

### 4.4. Fluorescent Properties and Carcinogen Identification

The fluorescent recognition properties of compound **I** were investigated as previously reported in the literature [2,7]. The sample was meticulously prepared as a 20 mg/L solution of methanol. Next, fully ground powder samples of **I** (2 mg) were immersed in solvents (5 mL) and subjected to ultrasonic treatment to prepare the stock suspension. The suspension was continuously stirred at a constant rate during fluorescence measurements to ensure homogeneity. The fluorescence intensities of **I** were recorded at room temperature upon excitation at 205 nm, within the wavelength range of 225–550 nm. The experiments were conducted in triplicate to ensure the reliability of the obtained data.

## 5. Conclusions

The successful synthesis of a tetra-Cd sandwiched trivacant Keggin-type tungstoarsenate [C_3_H_12_N_2_]_6_[Cd_4_Cl_2_(B-*α*-AsW_9_O_34_)_2_]·8H_2_O (C_3_H_10_N_2_ = 1,2-diaminopropane, abbreviated as **I**) was achieved through meticulous temperature control applied in this research. The novelty of the structure is that **I** is composed of two [B-*α*-AsW_9_O_34_]^9–^ fragments incorporating a remarkably rare {Cd_4_} cluster in the shape of a parallelepiped, which represents a pioneering example of a tetra-Cd cluster-substituted sandwich-type polyoxometalate (POM). Furthermore, it is noteworthy that the active site of the sandwiched cadmium cluster undergoes a substitution with a chlorine atom instead of the conventional oxygen. The novelty of both aforementioned structures is advantageous for compound **I** to showcase its exceptional fluorescence recognition characteristics.

## Figures and Tables

**Figure 1 molecules-29-05193-f001:**
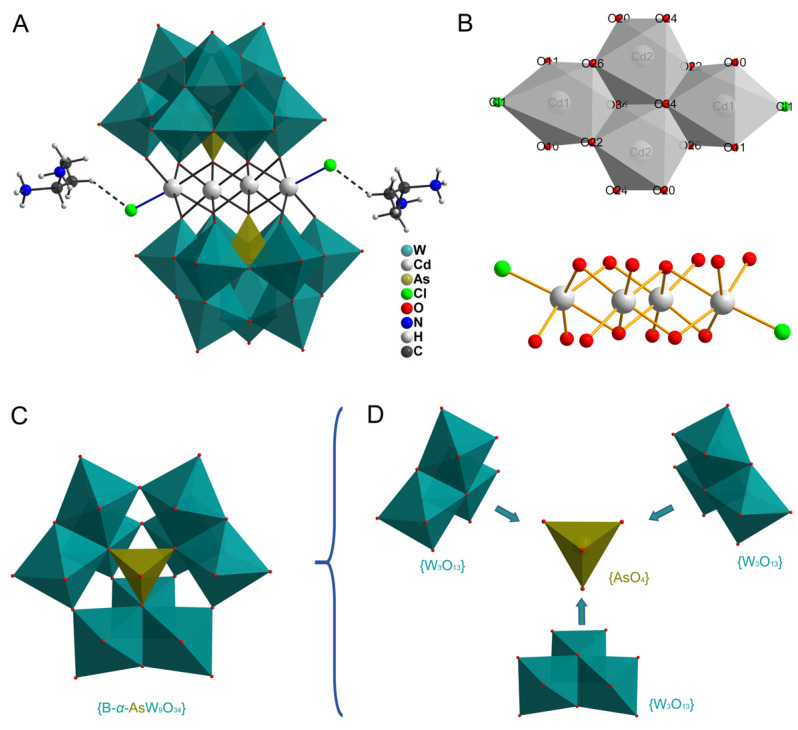
(**A**) Polyhedral and ball-and-stick representation of **I**. (**B**) Polyhedral or ball-and-stick representation of the tetra-Cd cluster {Cd_4_Cl_2_O_14_}. (**C**) Polyhedral representation of the trivacant Keggin-type fragment [B-*α*-AsW_9_O_34_]^9–^. (**D**) The anatomical view of the trivacant Keggin-type fragment [B-*α*-AsW_9_O_34_]^9–^.

**Figure 2 molecules-29-05193-f002:**
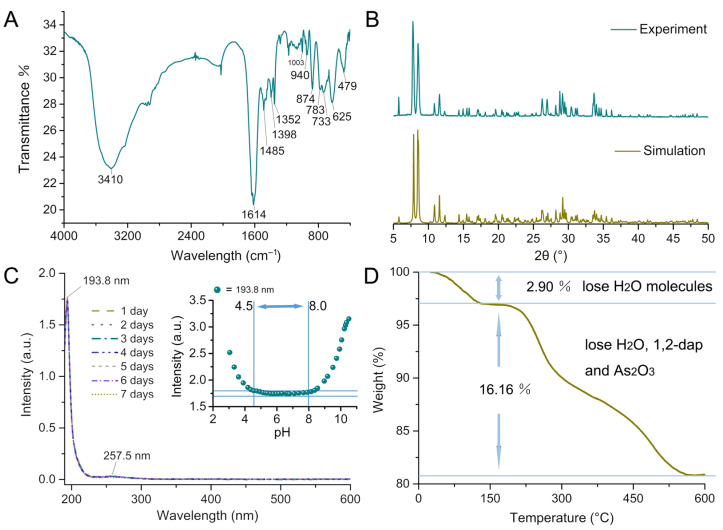
(**A**) IR spectrum for **I**; (**B**) comparison of the simulated and experimental XRPD patterns of **I**; (**C**) UV–Vis spectra of **I** in deionized water for 1–7 days and pH stability of **I** (inset figure in (**C**)); (**D**) TGA curve of **I**.

**Figure 3 molecules-29-05193-f003:**
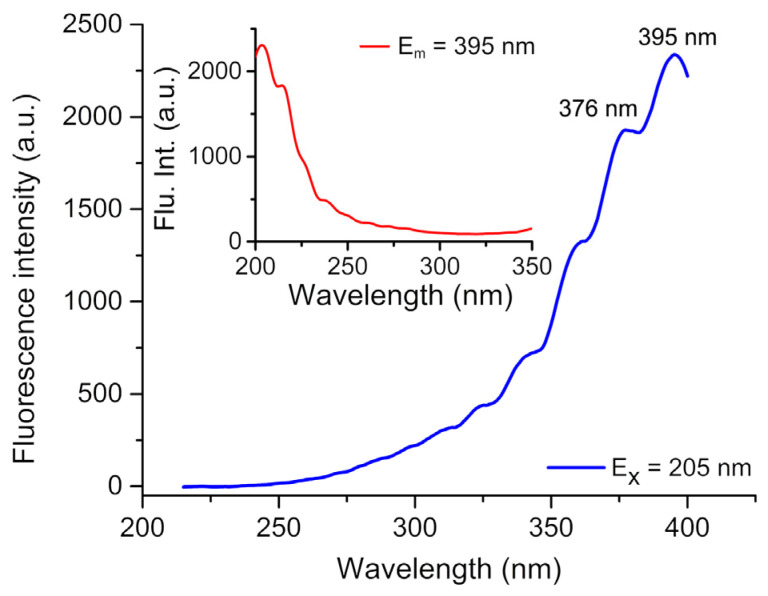
Emission spectrum and excitation spectrum (*λ*_ex_ = 205 nm) for compound **I** in the solid state at room temperature.

**Figure 4 molecules-29-05193-f004:**
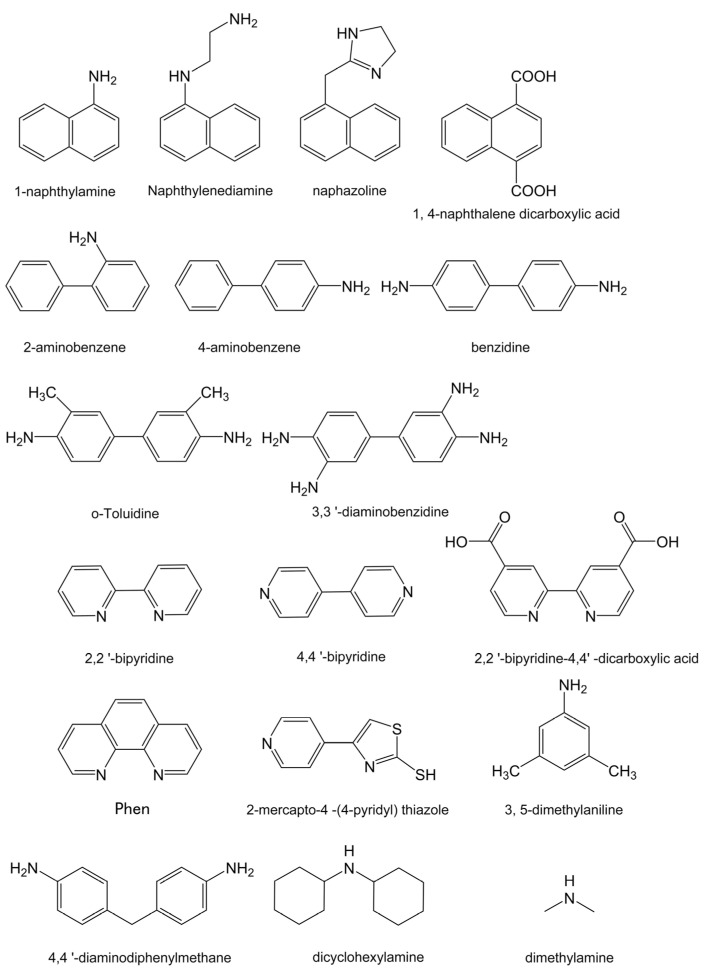
Diagrams of the structures of various carcinogenic dyes and amines used in parallel experiments.

**Figure 5 molecules-29-05193-f005:**
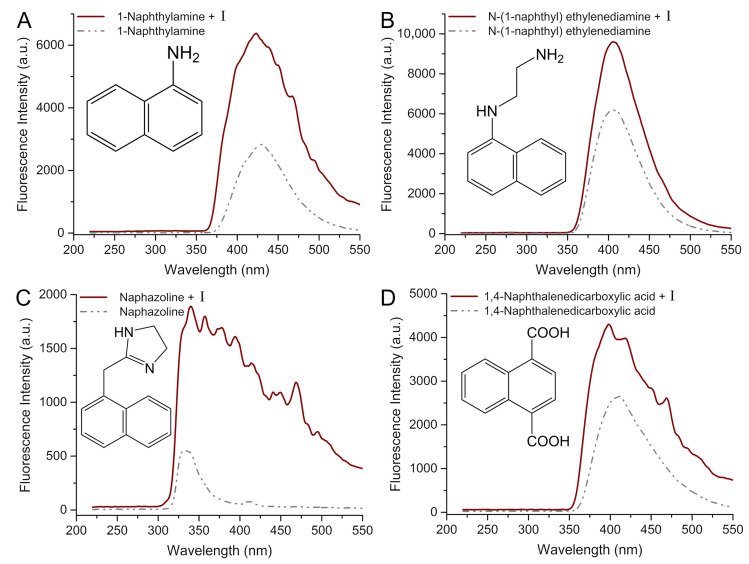
Comparison of fluorescence spectra with and without **I** (λ_ex_ = 205 nm) induced by 1-naphthylamine (**A**), N-(1-nphthyl)ethylenediamine (**B**), naphazoline (**C**), and 1,4-naphthalenedicarboxylic acid (**D**) in methanol.

**Figure 6 molecules-29-05193-f006:**
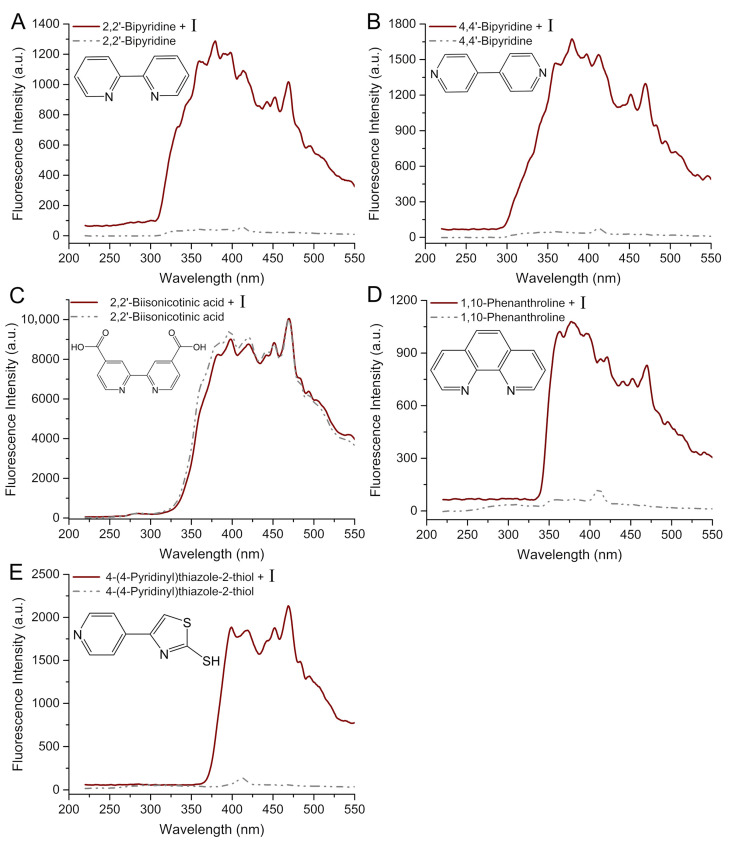
Comparison of fluorescence spectra with and without **I** (λ_ex_ = 205 nm) induced by 2,2′-bipyridine (**A**), 4,4′-bipyridine (**B**), 2,2′-biisonicotinic acid (**C**), 1,10-phenanthroline (**D**), and 4-(4-pyridinyl)thiazole-2-thiol (**E**) in methanol.

**Figure 7 molecules-29-05193-f007:**
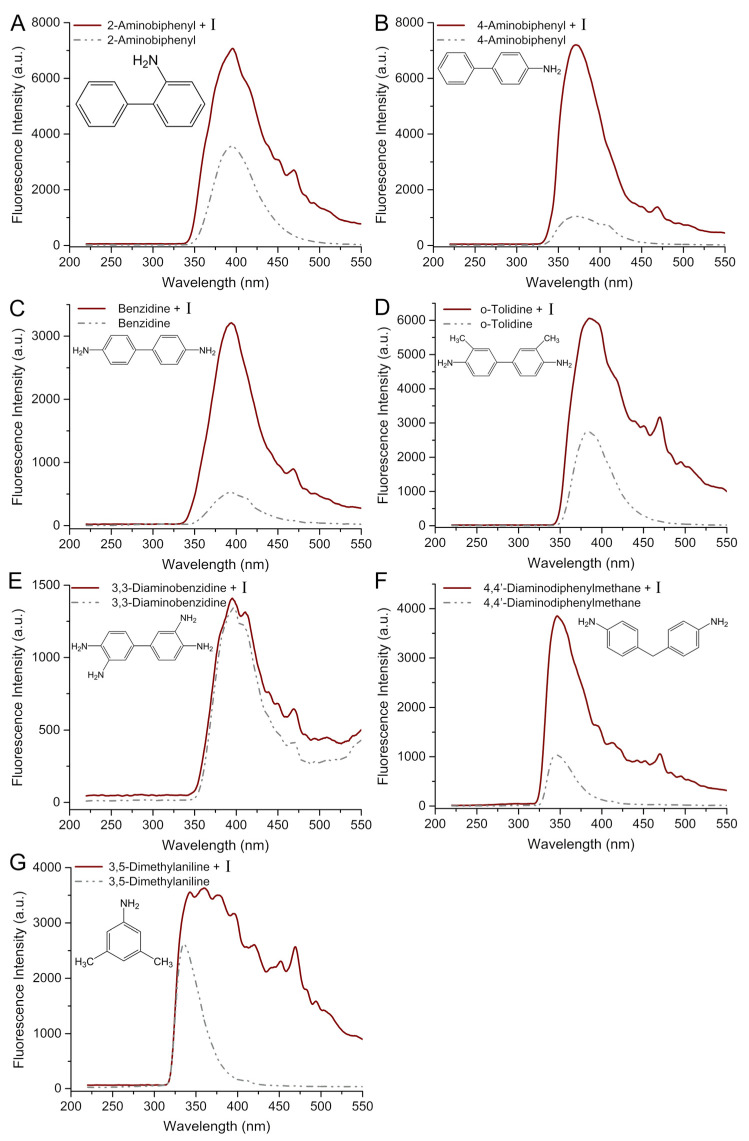
Comparison of fluorescence spectra with and without **I** (λ_ex_ = 205 nm) induced by 2-aminobiphenyl (**A**), 4-aminobiphenyl (**B**), benzidine (**C**), o-tolidine (**D**), 3,3-diaminobenzidine (**E**), 4,4-diaminodiphenylmethane (**F**), and 3,5-dimethylaniline (**G**) in methanol.

## Data Availability

Crystallographic data for the structural analysis have been deposited with the Cambridge Crystallographic Data Center, CCDC reference number 2341526, for Compound I. These data can be obtained free of charge from the Cambridge Crystallographic Data Center via www.ccdc.cam.ac.uk/data_request/cif 19 March 2024.

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
