# Peer review of "A d10-Cd Cluster Containing Sandwich-Type Arsenotungstate Exhibiting Fluorescent Recognition of Carcinogenic Dye in Methanol"

_molecules, 2024, doi:10.3390/molecules29215193_

Round 1

Reviewer 1 Report

Comments and Suggestions for Authors

Comment

Hua et al demonstrated a d10-Cd cluster containing sandwich-type arsenotungstate as fluorescent sensor for the recognition of carcinogenic dye in methanol. The investigation is interesting and the manuscript is well-written. It is suggested to accept after minor revision.

1.      Does compound 1 dissolve in methanol?

2.      What about the structural stability of compound 1 after the fluorescent sensing?

3.      In fluorescence recognition section, there are distinct enhanced fluorescence signals, how to determine what kind of dye cause the increase of signal if more dyes are in the solution.

4.      POMs usually undergo fluorescence quenching, why compound 1 can still emit fluorescence?

5.      How the Cl sites in compound 1 interact with dye molecules?

Author Response

Reviewer: #1

Thank you for your careful reviews and positive comments. We have revised our manuscript according to the suggestions.

Comment (C) 1: Does compound 1 dissolve in methanol?

Answer (A) 1: Thank you for your question. In the experiment, compound I exhibited insolubility in methanol; however, among the range of solvents investigated including methanol, isopropanol, ethanol, butan-1-ol, acetonitrile, ethane-1,2-diol, N-methylpyrrolidone, N,N-dimethylacetamide, tetrahydrofuran, N,N-Dimethylformamide, and chloroform, compound I demonstrated superior dispersibility specifically in methanol.

Table R1 Formula, structure, MW, logP, and polar surface area (PSA) of the solvent selected

Name

Formula

Structure

MW

log Pa

PSAa

1. methanol

(MeOH)

CH3OH

32.03

– 0.27

20.23

2. isopropanol

(2-PA)

CH3CHOHCH3

60.10

0.38

20.23

3. water

H2O

18.01

31.5

4. Ethanol

(EtOH)

CH3CH2OH

46.04

0.07

20.23

5. butan-1-ol

C4H9OH

74.07

0.97

20.23

6. acetonitrile

CH3CN

41.03

0.17

23.79

7. ethane-1,2-diol

HOCH2CH2OH

62.04

– 0.79

40.46

8. N-methylpyrrolidone

(NMP)

C3H6CONCH3

99.07

– 0.34

20.31

9. N,N-dimethylacetamide

(DMA)

N(CH3)2COCH3

87.07

– 0.49

20.31

10. acetone

CH3COCH3

58.04

0.2

17.07

11. tetrahydrofuran

(THF)

C4H8O

72.06

0.4

9.23

12. N,N-Dimethylformamide

(DMF)

C3H7NO

73.10

– 0.60

20.31

13. dichloromethane

CH2Cl2

83.95

1.01

0

14. chloroform

CHCl3

117.91

1.67

0

C2: What about the structural stability of compound 1 after the fluorescent sensing?

A2: Thanks for your question. As shown in Fig.R1, through the utilization of centrifugal separation for the isolation of hatched samples, IR analysis demonstrates that the fundamental framework of compound I remains.

Fig. R1 IR comparative analysis of compound I pre- and post-incubation with dimethylamine in methanol.

C3: In fluorescence recognition section, there are distinct enhanced fluorescence signals, how to determine what kind of dye cause the increase of signal if more dyes are in the solution.

A3: Thanks a lot. Compound I can currently only identify different solutions with one single component. And the compound I currently cannot accurately identify a specific dye molecule in the presence of multiple dyes. This is a very good question, and we have also tried to solve it. To test complex solutions, the current approach may necessitate employing a greater variety of probes and subsequently identifying the components they contain by determining the intersection of characteristic peaks. Herein, we solely presents one instance of this particular compound I, while subsequent studies are also encompassing the design and testing of diverse compounds.

C4: POMs usually undergo fluorescence quenching, why compound 1 can still emit fluorescence?

A4: Thanks a lot. The quenching of fluorescence in POMs have been previously attributed to the presence of Cu2+ or building blocks with a W5+ or Mo5+. The Cd in this report adopts a d10 structure, while the W also exhibits a +6 oxidation state. Therefore, compound I will not quench fluorescence but instead enhance the fluorescent signal.

C5: How the Cl sites in compound 1 interact with dye molecules?

A5: Thank you for your question. As a sandwich-type compound, the Cl occupying site in Compound I is highly reactive and can be substituted by various atoms or groups. In previous reports, we have previously reported a zinc-containing sandwich-type POM[1]. As shown in Fig. R2, the nitrogen on ethylenediamine has the potential to occupy similar sites within the compound. Therefore, the d10 metal ions in the sandwich layer possess the capability to coordinate with dyes containing nitrogen.

Fig. R2 (a) Combined polyhedral/ball-and-stick representation of 1. Lattice water molecules are omitted for clarity. (b) Ball-and-stick view of the [Zn(Hen)]618+ group. (c) Polyhedral view of the [B-α-AsW9O33]9 fragment. (d) The coordination environment around the ZnII cation.Color code: WO6 octahedra: green; As: teal; Zn: turquiose; N: blue; O: red; C: black; H: gray.

[1] Niu, J.Y.; Ma, X.; Zhao, J.W.; Ma, P.T.; Zhang, C.; Wang, J.P. A novel organic-inorganic hybrid turbine-shaped hexa-Zn sandwiched tungstoarsenate(III). CrystEngComm 2011, 13, 4834.

Reviewer 2 Report

Comments and Suggestions for Authors

This manuscript comes with some flowery language. As far as style, I would not object, but a few things need some attention.

1) Page 2, 'rhomblike' is not a scientific expression. One could suggest 'similar to a rhombus' or 'in the shape of a parallelepiped'.

2) Page 6, 'meticulously selected'. Sounds nice, but what does it mean? DoesFigure 4 represent a comprehensive list of molecules relevant for the purpose of the report? It continues with 'meticulously designed'. I think the correct expression is that all shown molecules are considered carzinogenic if at a concentration of at least 20mg/kg (kg of what?). I do not find all information needed. 

3) Page 11 'integration of dates' should be 'integration of data' unless you want to express something related to social interactions.

4) Table S1, space group P-1 should be written as P bar 1, use P followed by symbol #96 (dec.), then type in 1.

I now want to concentrate on the X-ray structure of compound 1.

The checkcif printout for the structure (after requesting the cif file from the CCDC) indicates an A-Alert of oxygen O4W being too close to itself. This is obviously a disordered water, despite the fact that it is missing its two hydrogen atoms. Similarly, three more waters are without hydrogen atoms. To get those onto a oxygen, find electron density maxima close to a oxygen, called, say O1, and call those maxima (type 2 for hydrogen) H1O, H1P. Now add above the weighting scheme in the .res file DFIX 0.8 0.005 O1 H1O O1 H1P and in the next line DFIX 1.27 0.005 H1O H1P. If needed, turn this water model with further dfix commands to find proper donor-acceptor interactions. Furthermore, there are some almost non-positive definite displacement ellipsoids, for example N4. The sum formula is not corresponding to the actual refinement model, which will be fixed if the water get their protons. Finally, the cif-file does not contain the diffraction data. Thus, do the following:

5) Put O4W on half-site occupancy.

6) Add all missing protons as described.

7) Check for missed disorder in the solvents and / or help the displacement parameters of solvents with DELU / Isor or RIGU or SIMU cards.

8) Refine with the newest version of SHELXL (it's free) and leave the automatically inserted fcf data in the cif file. 

9) Create a checkcif report that says on top '... found embedded fcf data ...'

10) Add the checkcif report and mended *.cif file to the molecules author submission.

P`1

Author Response

Reviewer: #2

Thanks a lot for your careful reviews and positive comments. We have revised our manuscript according to the suggestions.

Comment (C) 1: Page 2, 'rhomblike' is not a scientific expression. One could suggest 'similar to a rhombus' or 'in the shape of a parallelepiped'.

Answer (A) 1: Thank you for you suggestion. The sentence has been revised.

C2: Page 6, 'meticulously selected'. Sounds nice, but what does it mean? DoesFigure 4 represent a comprehensive list of molecules relevant for the purpose of the report? It continues with 'meticulously designed'. I think the correct expression is that all shown molecules are considered carzinogenic if at a concentration of at least 20mg/kg (kg of what?). I do not find all information needed.

A2: Thank you for your question. I sincerely apologize for any confusion caused by the imprecise wording in my previous statement. We initially chose these polycyclic aromatic hydrocarbons as representative compounds due to their prevalent presence in industrial products. The 'kg' in this unit should be changed to 'L', which refers to the volume of a solution.

C3: Page 11 'integration of dates' should be 'integration of data' unless you want to express something related to social interactions.

A3: Thanks a lot. It has been revised.

C4:  Table S1, space group P-1 should be written as P bar 1, use P followed by symbol #96 (dec.), then type in 1.

I now want to concentrate on the X-ray structure of compound 1.

The checkcif printout for the structure (after requesting the cif file from the CCDC) indicates an A-Alert of oxygen O4W being too close to itself. This is obviously a disordered water, despite the fact that it is missing its two hydrogen atoms. Similarly, three more waters are without hydrogen atoms. To get those onto a oxygen, find electron density maxima close to a oxygen, called, say O1, and call those maxima (type 2 for hydrogen) H1O, H1P. Now add above the weighting scheme in the .res file DFIX 0.8 0.005 O1 H1O O1 H1P and in the next line DFIX 1.27 0.005 H1O H1P. If needed, turn this water model with further dfix commands to find proper donor-acceptor interactions. Furthermore, there are some almost non-positive definite displacement ellipsoids, for example N4. The sum formula is not corresponding to the actual refinement model, which will be fixed if the water get their protons. Finally, the cif-file does not contain the diffraction data. Thus, do the following:

A4: Thank you for your suggestion. We have meticulously revised the data as suggestion. Please find the latest data package in the attachment, and non-published files

C5: Put O4W on half-site occupancy.

A5: Done.

C6: Add all missing protons as described.

A6: Done.

C7: Check for missed disorder in the solvents and / or help the displacement parameters of solvents with DELU / Isor or RIGU or SIMU cards.

A7: Done.

C8: Refine with the newest version of SHELXL (it's free) and leave the automatically inserted fcf data in the cif file.

A8: Done.

C9: Create a checkcif report that says on top '... found embedded fcf data ...'

A9: Done.

C10: Add the checkcif report and mended *.cif file to the molecules author submission.

A10: Done.

Round 2

Reviewer 2 Report

Comments and Suggestions for Authors

The authors have acted on the points I made. It maybe me, but I could not detect a checkcif pdf file in the submitted revision, so I only hope the given answers in their response are given in good faith.